# Insights into the recognition mechanism in the UBR box of UBR4 for its specific substrates

Da Eun Jeong[1,2], Hye Seon Lee[3], Bonsu Ku [3], Cheol-Hee Kim [2], Seung Jun Kim [1,4 ✉] & Ho-Chul Shin [1,5 ✉]

The N-end rule pathway is a proteolytic system involving the destabilization of N-terminal amino acids, known as N-degrons, which are recognized by N-recognins. Dysregulation of the N-end rule pathway results in the accumulation of undesired proteins, causing various diseases. The E3 ligases of the UBR subfamily recognize and degrade N-degrons through the ubiquitin-proteasome system. Herein, we investigated UBR4, which has a distinct mechanism for recognizing type-2 N-degrons. Structural analysis revealed that the UBR box of UBR4 differs from other UBR boxes in the N-degron binding sites. It recognizes type-2 N-terminal amino acids containing an aromatic ring and type-1 N-terminal arginine through two phenylalanines on its hydrophobic surface. We also characterized the binding mechanism for the second ligand residue. This is the report on the structural basis underlying the recognition of type-2 N-degrons by the UBR box with implications for understanding the N-end rule pathway.

[1] Critical Disease Diagnostics Convergence Research Center, Korea Research Institute of Bioscience and Biotechnology (KRIBB), Daejeon 34141, Republic of Korea. [2] Department of Bioscience & Biotechnology, Chungnam National University, Daejeon 34134, Republic of Korea. [3] Disease Target Structure Research Center, Division of Biomedical Research, KRIBB, Daejeon 34141, Republic of Korea. [4] Department of Proteome Structural Biology, KRIBB School of Bioscience, University of Science and Technology, Daejeon 34113, Republic of Korea. [5] Graduate School of New Drug Discovery and Development, Chungnam National University, Daejeon 34134, Republic of Korea. ✉email: ksj@kribb.re.kr; shinhc81@kribb.re.kr

The N-end rule pathway is a proteolytic system that degrades unnecessary, unfolded, and misfolded proteins by destabilizing the N-terminal amino acids (N-degrons) recognized by N-recognins[1–5]. This pathway regulates protein homeostasis by removing unnecessary proteins from many cellular signaling pathways, such as apoptosis[6–9], autophagy[10–12] and inflammatory signaling pathways[13–16]. Dysregulation of the N-end rule pathway leads to the accumulation of undesired proteins in the cellular environment and can induce many diseases, such as cancer[17–22], neurodegenerative disease[23–25], and Johanson–Blizzard syndrome[26–30].

N-degrons are classified as type-1, containing positively charged amino acids (Arg, Lys, and His), and type-2, containing bulky hydrophobic amino acids (Phe, Tyr, Trp, Leu, and Ile)[31]. Various types of N-degrons are recognized by specific N-recognins and degraded through the ubiquitin-proteasome system (UPS)[32] or autophagy[10,11,33].

The UPS attaches ubiquitin, a 76-amino acid polypeptide, to a substrate as a tag for degradation[34,35]. Substrate ubiquitination occurs through a series of enzymes, including ubiquitin-activating enzymes (E1s), ubiquitin-conjugating enzymes (E2s), and E3 ubiquitin ligases (E3s). E1 enzymes are activating enzymes that activate ubiquitin and transfer it to E2 enzymes. E2 enzymes are conjugating enzymes that obtain ubiquitin from E1 enzymes and transfer it to E3 ubiquitin ligases[33,36–40]. E3 ubiquitin ligases are key enzymes that recognize specific substrates and attach ubiquitin tags, marking them for rapid degradation by the proteasome. Because most N-recognins belong to the family of E3 ligases, it is important to identify the interaction between N-recognins and their substrates with destabilizing N-degrons to understand the N-end rule pathway.

UBR proteins (UBR1–UBR7) are a subfamily of E3 ligases that recognize substrates and mediate their degradation through the UPS[3,41,42]. These UBR proteins share UBR boxes, which are domains of approximately 70 amino acids that coordinate with three zinc ions[33,43–45].

According to a previous study, type-1 and type-2 N-degrons are recognized by different domains in UBR proteins[46,47]. Type-1 N-degrons are recognized by UBR boxes, whereas type-2 N-degrons are recognized by N-domains, which are present only in UBR1 and UBR2[48]. UBR1 and UBR2 recognize type-1 N-terminal arginine through their UBR box domain and type-2 N-terminal phenylalanine through their N-domain[48]. UBR5 cannot bind to type-2 N-terminal phenylalanine because it only has a UBR box domain[48]. However, unlike other UBR proteins, UBR4, despite having only the UBR box domain, can recognize both type-1 and type-2 N-degron[48].

UBR4, a member of the UBR subfamily, is implicated in various biological processes, such as yolk sac vascular development[49], neurogenesis[50,51], neuronal survival[50,51], and myofiber size determination[52,53]. A previous study showed that UBR4 knockout in mice resulted in early embryonic lethality[50]. However, the molecular mechanism underlying the recognition of UBR4 of N-degrons remains unknown.

Although UBR4 has only a UBR box domain, it can bind to both type-1 arginine and type-2 phenylalanine[48]. To investigate the specificity of the UBR box structure of UBR4, we determined the apo- and ligand-bound structures of the UBR box. These structures demonstrate that the UBR box of UBR4 differs from other UBR boxes in the N-degron binding sites and provide insight into the mechanism of substrate binding. We also performed thermal shift assays (TSA) and isothermal titration calorimetry (ITC) to investigate the binding of tetrapeptides with different amino acids at their first and second positions to the UBR box of UBR4. Our findings suggest that the UBR box of UBR4 recognizes type-1 N-terminal arginine and type-2 N-terminal amino acids containing an aromatic ring, such as tyrosine, tryptophan, and phenylalanine through pi-interaction with two phenylalanines of the UBR box of UBR4.

## Results

### The UBR box of UBR4 recognizes both type-1 and type-2 N-degrons.

The UBR box, a common feature among UBR family proteins, can recognize type-1 N-degron[44] and the N-domain that is only in UBR1 and UBR2 can recognize type-2 N-degron[48]. However, UBR4 recognizes both type-1 and type-2 N-degrons without the N-domain[48]. The affinity between the UBR box of UBR4 (1669–1729, UBR4$^{UBR}$) and its ligands was measured using ITC to determine whether UBR4$^{UBR}$ directly recognizes both types of N-degrons, or if there is another unknown domain involved in recognition. We used the following three types of tetrapeptides: RIFS, a type-1 N-degron, which is often used as a substrate for the UBR box; YIFS, a type-2 N-degron, in which only the N-terminal residue of the peptide is changed; and VIFS, which served as a negative control (Fig. 1a). Because of low binding affinity, the correct calculation of the dissociation constant ($K_D$) value was difficult; however, based on the Kcal/mole of injectant value related to enthalpy change ($\Delta H$), we inferred that YIFS would have more stable binding than RIFS, given that the value for YIFS was approximately 2.5 times larger than that for RIFS (Fig. 1a). For accurate measurement of binding affinity, we conducted experiments with the highest possible concentration of the samples. Both proteins and substrates were tested at concentrations 4.5 times higher than the standard concentrations. The results of the binding affinity measurements revealed that RIFS exhibited a binding affinity, $K_D = 626 \pm 29$ μM, while YIFS showed approximately 1.6 times higher binding affinity than RIFS with $K_D = 385 \pm 60$ μM (Fig. 1b, c). Furthermore, VIFS did not show any binding affinity (Fig. 1b, c). In conclusion, we found that UBR4$^{UBR}$ can directly recognize type-1 and type-2 N-degrons but YIFS and RIFS are not real substrate sequences for UBR4.

### UBR4 exhibits different ligand selectivity compared with other UBRs.

The binding affinities of the tetrapeptides RIFS and YIFS to UBR4$^{UBR}$ were observed to be significantly lower than that between the UBR box of UBR2 and RIFS, reported in previous studies ($K_D = \sim 2$ μM)[44,54]. UBR boxes have two pockets that recognize the first and second residues of the ligand[33,55,56]. Furthermore, there is a significant difference in their binding affinity, which depends not only on the first residue of the ligand but also on the type of the second amino acid residue[33]. Therefore, we inferred that the influence of the second residue may cause the low binding affinity between UBR4$^{UBR}$ and YIFS, and between UBR4$^{UBR}$ and RIFS.

To investigate the amino acid sequence of the ligand preferred by UBR4$^{UBR}$, we performed TSA on 20 species of YXFS with UBR4$^{UBR}$, where X represents the 20 natural amino acids[54,57–59]. The melting curve of TSA on 20 species of YXFS with UBR4$^{UBR}$ has shown in Supplementary Fig. 3. According to the TSA results, small residues, such as those of alanine, serine, glycine, and cysteine, showed a greater change in the melting temperature compared to control without the addition of peptides or proteins ($\Delta Tm$) (Fig. 2a). Additionally, longer residues, such as glutamate and methionine, also exhibited a higher $\Delta Tm$ (Fig. 2a). The maximum increase in the melting temperature was approximately 12 °C for alanine, followed by approximately 10 °C for glutamate. In contrast, the smallest increases were observed for phenylalanine (~3 °C), arginine (~3 °C), histidine (~3 °C), and aspartate (~3 °C), indicating weak or no binding (Fig. 2a). Interestingly, a highly specific observation was that YEFS exhibited the second-

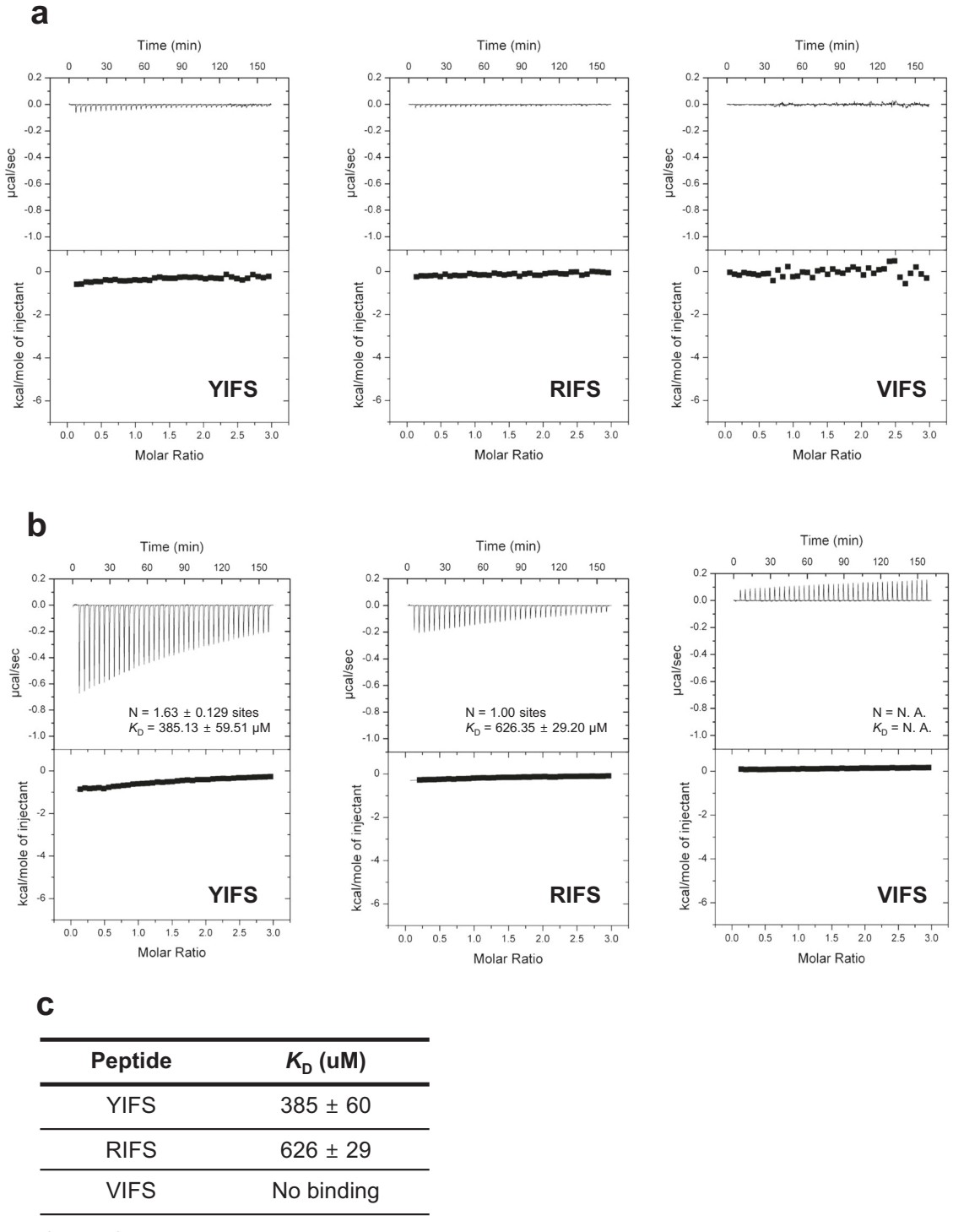

**Fig. 1 Isothermal titration calorimetry (ITC) data for UBR4$^{UBR}$ and three ligands: type-2 N-degron, YIFS. type-1 N-degron, RIFS. Negative control, VIFS. a** 0.05 mM of UBR4$^{UBR}$ and 1 mM of peptide **b** The concentrations of protein and peptide showed 4.5 times higher than the general condition in our ITC assay, 0.225 mM of UBR4$^{UBR}$ and 4.5 mM of peptide. **c** The dissociation constant ($K_D$) values measured through isothermal titration calorimetry for three peptides with UBR4$^{UBR}$.

highest binding affinity for UBR4$^{UBR}$, whereas YDFS exhibited very weak binding affinity (Fig. 2a). This is because glutamate and aspartate have the same carboxyl group but differ in length by just one carbon atom.

To verify whether the relative affinity obtained through ΔTm measurements remained consistent, we performed ITC to determine the $K_D$ for the binding of UBR4$^{UBR}$ with YAFS, YEFS, and YDFS. For YAFS and YEFS the $K_D$ values were 18.9 ± 1.2 and 18.3 ± 0.9 μM, respectively (Fig. 2b and Supplementary Fig. 1a). These values were nearly identical within the margin of error. However, similar to the comparison of ΔTm values, no detectable binding affinity was observed for YDFS (Fig. 2b and Supplementary Fig. 1a).

**a**

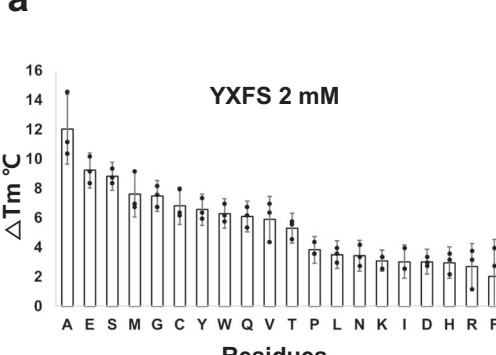
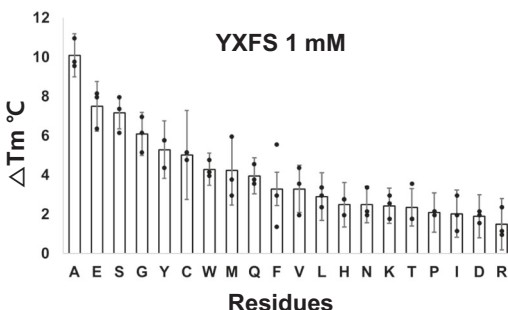

**b**

| Peptide | $K_D$ (uM) |
|---------|------------|
| YAFS | 18.9 ± 1.2 |
| YEFS | 18.3 ± 0.9 |
| YDFS | No binding |

\* Limit of detection : 50uM

**c**

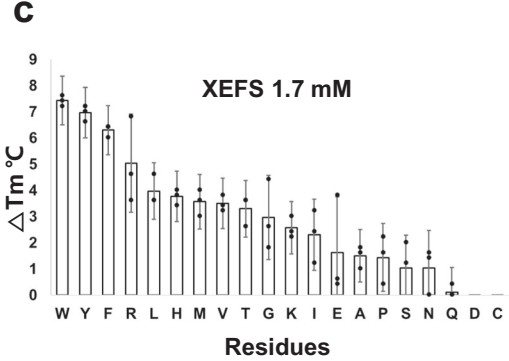
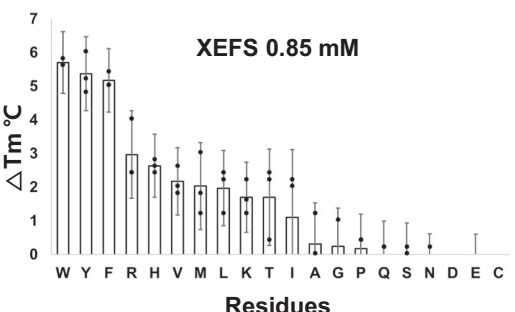

**d**

| Peptide | $K_D$ (uM) |
|---------|------------|
| YEFS | 18.3 ± 1.2 |
| WEFS | 21.2 ± 5.1 |
| REFS | 43.6 ± 17.0 |
| HEFS | 55.8 ± 3.5 |
| DEFS | No binding |

\* Limit of detection : 50uM

**Fig. 2 TSA and ITC data to confirm the first and second amino acid residues with the highest binding affinity to the UBR4[UBR]. a** Results of thermal shift assay (TSA) for 20 different ligands (YXFS) at concentrations of 2 and 1 mM. The results are sorted in descending order of ΔTm. **b** The dissociation constant ($K_D$) values measured using isothermal titration calorimetry (ITC) for the three peptides with UBR4[UBR]. **c** Results of TSA for 20 different ligands (XEFS) at concentrations of 1.7 and 0.85 mM. The results are sorted in descending order of ΔTm. **d** The ITC was performed to determine the $K_D$ values for the five peptides with UBR4[UBR].

To determine whether the inability to calculate the binding affinity for YDFS was due to its complete absence of binding or its weak affinity, we performed ITC assay at a 4.5 times higher concentration. The calculated $K_D$ value for YDFS was $124 \pm 6$ μM (Supplementary Fig. 1b). This result revealed that YDFS has a low binding affinity to UBR4$^{UBR}$. However, this low affinity does not have biological significance, and thus, this peptide is not a suitable ligand.

Based on the differences in the binding affinity for various second amino acids, we further assessed the differences in binding affinities corresponding to different N-terminal amino acids. Twenty XEFS peptides were selected based on the affinity measurements for the second amino acid variation. Because the introduction of a hydrophobic residue at the first position may lead to solubility problems with XAFS, we chose XEFS instead of XAFS.

Based on TSA results, type-2 ligands with an aromatic ring induced high ΔTm values for UBR4$^{UBR}$. Additionally, although type-1 ligands showed relatively high binding affinity, the affinity was lower than that for type-2 ligands with an aromatic ring (Fig. 2c). The maximum increase in Tm was approximately 7.5 °C for tryptophan, followed by approximately 7 °C for tyrosine. Arginine was also observed to induce a considerable increase of approximately 5 °C in Tm (Fig. 2c). The melting curve of TSA on 20 species of XEFS with UBR4$^{UBR}$ has shown in Supplementary Fig. 4.

ITC was performed to determine the $K_D$ for UBR4$^{UBR}$ and the selected XEFS peptides. YEFS exhibited the highest affinity ($K_D = 18.3 \pm 1.2$ μM), followed by WEFS ($K_D = 21.2 \pm 5.1$ μM) although the results were almost indistinguishable considering the margin of error (Fig. 2d and Supplementary Fig. 2a, b). In comparison, type-1 peptides showed a lower affinity than these two peptides. The $K_D$ for REFS and HEFS were $43.6 \pm 17.0$ and $55.8 \pm 3.5$ μM, respectively (Fig. 2d and Supplementary Fig. 2c, d). DEFS showed no detectable binding affinity (Fig. 2d and Supplementary Fig. 2e).

**Identification of two subfamilies of the UBR box based on the crystal structure of UBR4$^{UBR}$ and sequence alignment of the UBR boxes of the UBR family**. Unlike other UBR boxes, UBR4$^{UBR}$ prefers ligands in which those N-terminal residue has an aromatic ring (Fig. 2c). To provide a basis for type-2 recognition, we determined the structure of UBR4$^{UBR}$.

First, we determined the ligand-free structure of UBR4$^{UBR}$. The structure of UBR4$^{UBR}$, like other UBR boxes, is composed of three α-helices and two antiparallel β-sheets (Fig. 3a). To assess the differences between UBR4$^{UBR}$ and other canonical UBR boxes, we aligned the crystal structure of UBR4$^{UBR}$ with the UBR boxes of UBR1 and UBR2. The UBR boxes of Human UBR1 and Human UBR2 showed similar structures, with a root-mean-square deviation (RMSD) value of 0.363 (Fig. 3b). The UBR box of UBR1 extracted from full-length structure (PDB ID: 7MEX) and the UBR box of Human UBR1 also exhibited similar structures, the RMSD value of 1.521 (Supplementary Fig. 5)[45]. However, the RMSD value between Human UBR1 and Human UBR4 was 2.227, and that between Human UBR2 and Human UBR4 was 3.314 (Fig. 3b). These results indicate that UBR4$^{UBR}$ is different from the UBR boxes of UBR1 and UBR2. Furthermore, we aligned UBR4$^{UBR}$ with the UBR box domain of UBR1 from *Saccharomyces cerevisiae* to determine if UBR4$^{UBR}$ showed resemblances to the canonical UBR boxes of other species. Although the RMSD value between Yeast UBR1 and Human UBR1 was 0.980 and between Yeast UBR1 and Human UBR2 was 0.597, the RMSD value between Yeast UBR1 and Human UBR4 was 4.282, indicating that Human UBR4 exhibits less similarity with UBR1 and UBR2 even with other species (Fig. 3b).

The UBR4$^{UBR}$ coordinates three zinc ions (Fig. 3c). One zinc ion (Zn1) is canonically coordinated by two cysteines and two histidines similar to the other UBR boxes (Fig. 3c). However, whereas two zinc ions (Zn2 and Zn3) were coordinated by six cysteines and one histidine in the structure of the UBR boxes of UBR1 and UBR2, two zinc ions were coordinated by seven cysteines in UBR4$^{UBR}$ (Fig. 3c). In UBR4$^{UBR}$, unlike in the UBR boxes of UBR1 and UBR2, His166, which coordinates Zn2, was absent. Instead of a histidine residue, Cys1724 was coordinated with Zn2 (Fig. 3c). Moreover, based on the alignment of amino acid sequences of the seven UBR boxes, it was confirmed that UBR1, UBR2, and UBR3 possess a histidine residue that coordinates Zn2 (Fig. 3d, red box). In contrast, in UBR4, UBR5, UBR6, and UBR7, a cysteine residue was conserved instead of histidine in the protein sequence (Fig. 3d, orange box). Based on the results of this analysis, UBR family proteins can be classified into two subfamilies—subfamily-1, including UBR1, and subfamily-2 including UBR4—according to their zinc-coordinating residues.

**Aspartate that gives preference to arginine is not structurally conserved in UBR4**. Besides differences in the residues coordinating zinc, there was also a structural difference between UBR4$^{UBR}$ and the UBR boxes of UBR1 and UBR2. When bound to the UBR box of UBR2, the amino group at the N-terminus of the ligand had a strong binding affinity through hydrogen bonding with the carboxyl group of the side chain of Asp150 and the oxygen atoms of the peptide bonds between Phe148 (Fig. 4a)[60]. In UBR1 and UBR2, Asp153 is an important residue that enhances the binding affinity for arginine and provides residue selectivity by forming a hydrogen bond with the positively charged guanidino group of the arginine residue in the ligand[60] (Fig. 4a).

The carboxyl group of Asp1715 and the oxygen atom between Phe1713 were conserved at the same positions in the structure of UBR4$^{UBR}$ as in the UBR box of UBR2 (Fig. 4b)[60]. However, in the structure of UBR4$^{UBR}$, there is a unique feature where the aspartic acid residue (Asp1721), which is one of the main features of N-recognins such as UBR1 and UBR2, is located far away from the ligand-binding site (Fig. 4b, c). In other words, in UBR4$^{UBR}$, Asp1721 may not be involved in ligand-binding unlike other the UBR boxes of UBR1 and UBR2.

**Selective binding between the ligand and UBR4$^{UBR}$ is mediated by two phenylalanines**. We initially hypothesized that Asp1721 is critical for ligand binding. However, in the tertiary structure, Asp1721 was distant from the ligand-binding site and appeared to be unrelated to ligand binding. Therefore, we attempted to determine the structure of UBR4$^{UBR}$ in complex with YIFS to investigate how the distinctive feature of UBR4$^{UBR}$ observed in its apo structure affects ligand binding and how UBR4$^{UBR}$ can recognize type-2 N-degrons.

First, as observed in other N-recognin structures, during crystallization of UBR4$^{UBR}$ the amino acid at the N-terminus was bound to the ligand-binding site of other UBR4$^{UBR}$ molecules in the apo UBR4$^{UBR}$ structure. Furthermore, even when complexed with a peptide ligand, this phenomenon occasionally causes ligand dissociation during crystallization. To overcome this issue, we purified the protein with the N-terminus starting with tyrosine (YIFS-UBR4$^{UBR}$) using the LC3B tag system. In a previous study, the crystal structure of N-recognins with desired sequences of ligands, such as RLGS-yUBR1 UBR box and GEEED-p62 ZZ domain, were determined through the LC3B tag system.

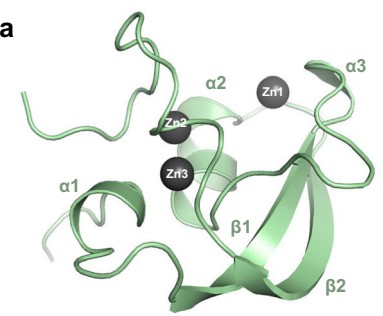

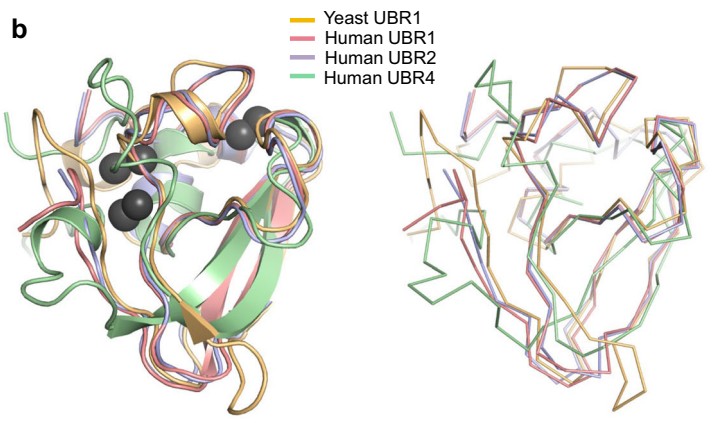

| Protein 1 | Protein 2 | RMSD value | Protein 1 | Protein 2 | RMSD value |
|---|---|---|---|---|---|
| Human UBR1 | Human UBR2 | 0.363 (54 to 54 atoms) | Yeast UBR1 | Human UBR1 | 0.980 (45 to 45 atoms) |
| Human UBR1 | Human UBR4 | 2.227 (43 to 43 atoms) | Yeast UBR1 | Human UBR2 | 0.597 (45 to 45 atoms) |
| Human UBR2 | Human UBR4 | 3.314 (30 to 30 atoms) | Yeast UBR1 | Human UBR4 | 4.282 (39 to 39 atoms) |

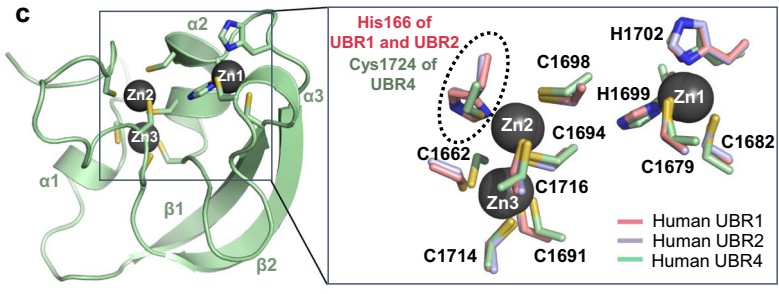

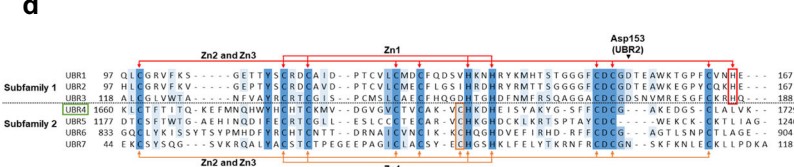

The LC3B tag consists of C-terminal glycine; the ATG4B protease can cleave the peptide bond after the C-terminal glycine of the LC3B tag. After cleavage of the LC3B tag, the desired N-terminal residue which can be any 20 amino acids except for proline is exposed at the start of the target substrate (Supplementary Fig. 6a). By using this system, we can yield a target protein with the desired N-terminal residue upon cleavage with ATG4B protease[61]. Using this system we successfully established the crystal structure of YIFS-UBR4[UBR] (Fig. 5a and Supplementary Fig. 6b). We found that the N-terminal residue of one YIFS-UBR4[UBR] molecule binds to the ligand-binding site of another (Supplementary Fig. 6b). The structural reliability of YIFS residues was further confirmed through the omit map (Supplementary Fig. 6c).

**Fig. 3 Subfamily identification through the structural comparison and amino acid sequence analysis between the UBR4$^{UBR}$ and various UBR boxes.**
**a** Overall structure of UBR4$^{UBR}$ is represented in the cartoon and ball format. **b** Crystal structure of Human UBR4$^{UBR}$ aligned with the UBR box of Human UBR1, Yeast UBR1, and Human UBR2. The aligned structure is represented in two ways, using cartoon and ribbon. Root-mean square deviation (RMSD) values of C atoms for aligned structures of four UBR boxes are also represented in the table. **c** The positions of cysteine and histidine residues that coordinate zinc ions in UBR4$^{UBR}$ compared with those in UBR1 and UBR2. **d** The amino acid sequences of the seven UBR domains are aligned and residues coordinating zinc ions are indicated. The differences between the two subfamilies are indicated by the presence of histidine (H) and cysteine (C) residues, coordinating Zn2 (red and orange box, respectively). The Asp153 residue of UBR2 is located in the arginine recognition loop and recognizes the N-terminal arginine residue of the ligand. The Yeast UBR1, Human UBR1, Human UBR2, and Human UBR4 represented different colors as lightorange, salmon, lightblue, and palegreen, respectively.

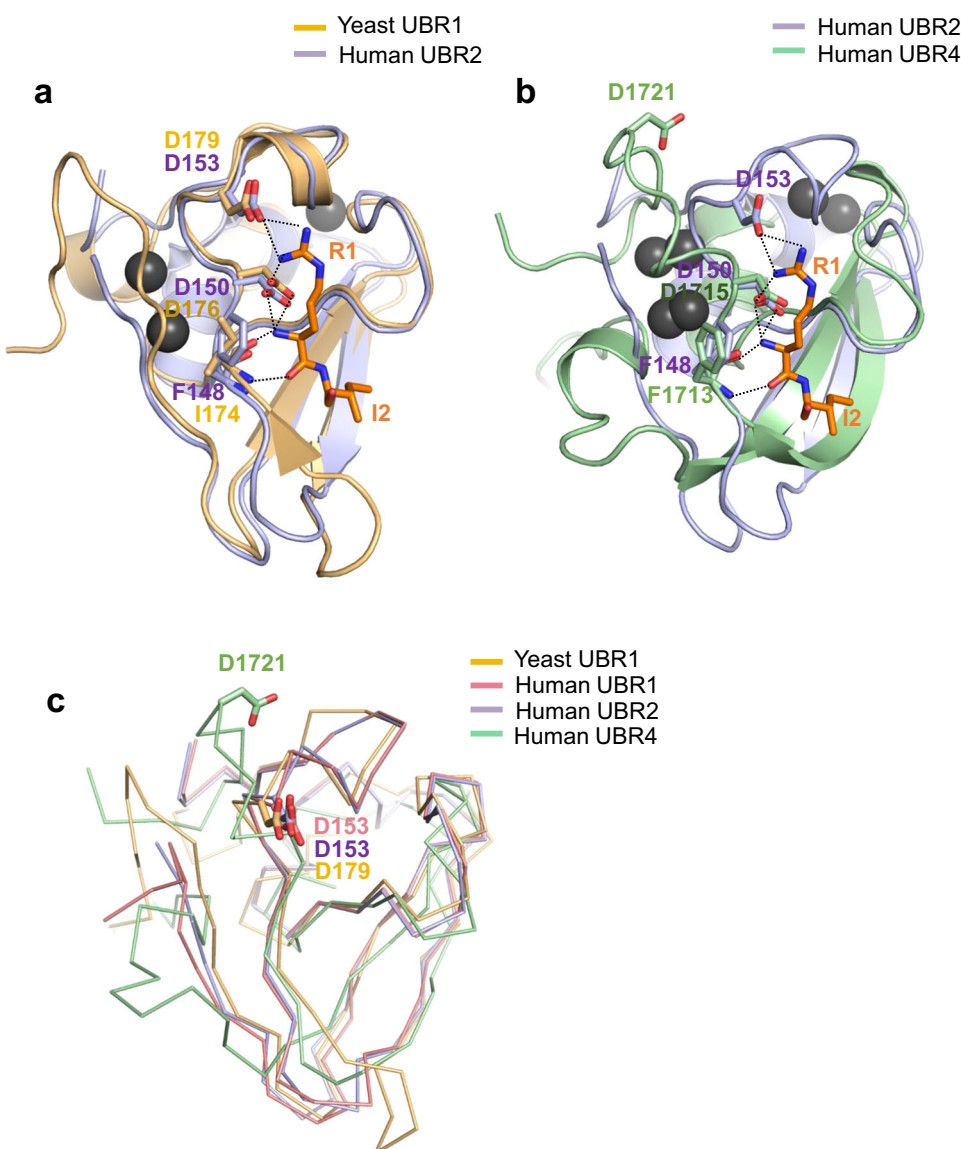

**Fig. 4 Structural comparative analysis of ligand-binding sites. a** Crystal structure of the Human UBR2–RIFS complex (PDB ID: 3NY3, lightblue) aligned with Yeast UBR1 (PDB ID : 3NIH, lightorange) and shown in cartoon, ball, and stick representations. Residues of UBR2 that bind to the first amino group and Asp153, which recognizes the guanidino group of arginine residue of the ligand, are shown. **b** The apo structure of Human UBR4$^{UBR}$ (palegreen) is compared with that of the Human UBR box of UBR2. Residues of UBR4 corresponding to those of UBR2 that bind to the ligand are indicated. Hydrogen bonds are indicated by dashed lines. **c** The structures of the UBR boxes are compared to indicate the location of the arginine recognition loop. The aspartate residue that recognizes the N-terminal arginine of the ligand is shown in the stick representation.

As expected, the first amino group of YIFS-UBR4$^{UBR}$ was bound to the carboxyl group of Asp1715 and the oxygen atom of Phe1713, as in the other UBR boxes. However, unlike the structure of the arginine-bound UBR box of UBR1 and UBR2, the direction of the N-terminal tyrosine residue was not toward

Asp1721, but toward Phe1671 and Phe1713 in the binding site (Fig. 5a). These two phenylalanine residues on the ligand-binding surface of UBR4$^{UBR}$ create a hydrophobic patch, which explains the binding ability of residues with aromatic rings. However, this does not explain the low binding affinity of other hydrophobic

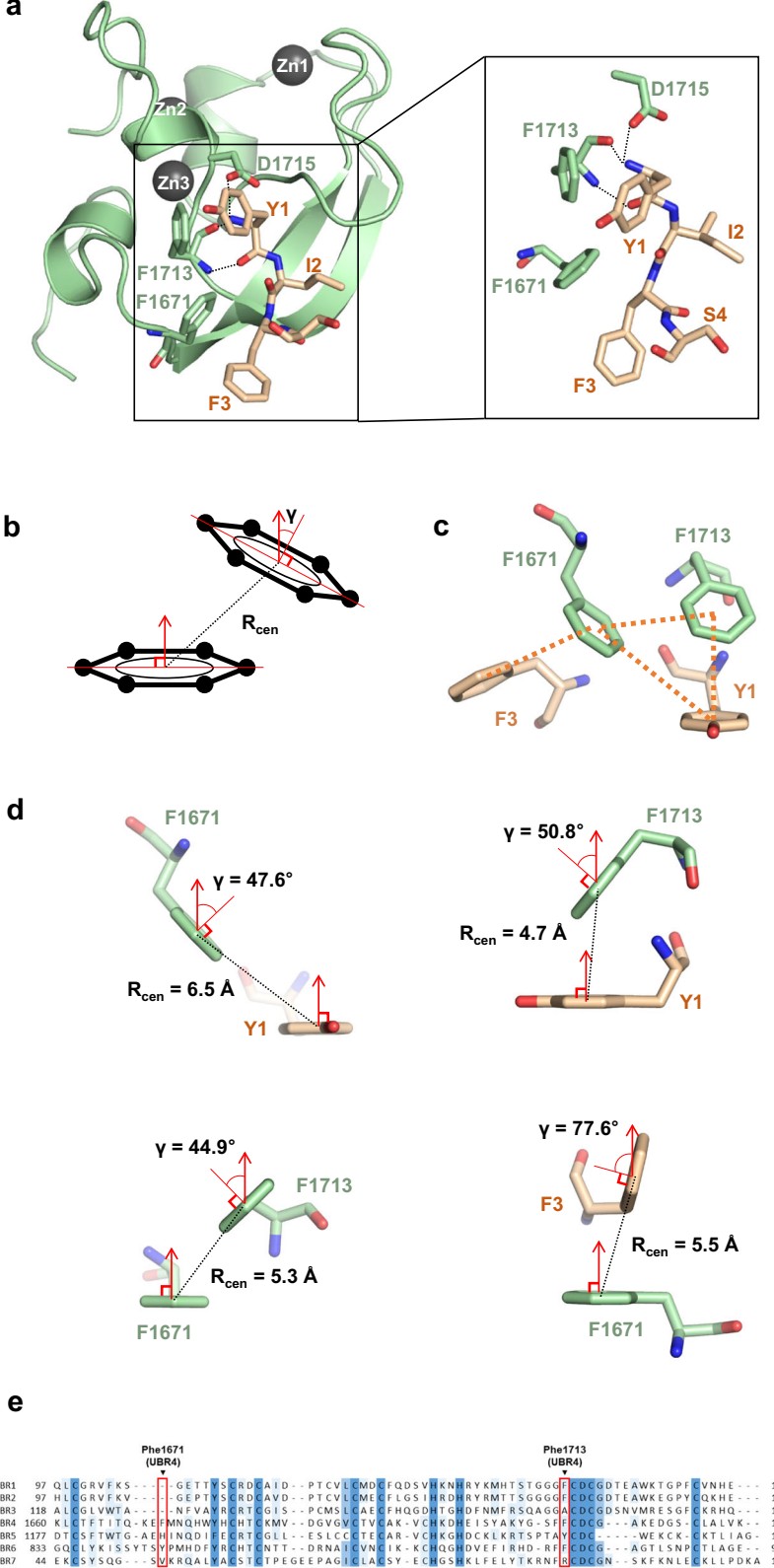

**Fig. 5 The YIFS-UBR4^UBR structure and analysis of the binding site of the first residue. a** Structure of UBR4^UBR bound to the ligand YIFS revealed through the YIFS-UBR4^UBR structure determination. Residues that bind to the first amino group of the ligand are shown in stick representation; Tyr1 of the ligand is facing phenylalanines of UBR4. The dashed lines indicate hydrogen bond. **b** A diagram explaining the distance between the centers of the aromatic rings ($R_{cen}$) and the angle ($\gamma$) between the perpendicular surface normal. **c** Pi-stacking interactions between the two phenylalanine residues (Phe1671 and Phe1713) of UBR4 and Tyr1 and Phe3 residues of the ligand in stick representation. **d** The figure displays the $R_{cen}$ and $\gamma$ values for each pi-interaction. **e** The residues corresponding to F1671 and F1713 of UBR4^UBR are highlighted as two red boxes on the aligned amino acid sequences of the seven UBR domains.

residues or the ability of arginine to bind. To comprehensively investigate the binding mechanism of the N-terminal ligand-binding residue, we introduced various mutations into UBR4$^{UBR}$ and performed ITC with tetrapeptides.

Based on the alignment of the crystal structures of UBR boxes of UBR1, UBR2, and UBR4 and the ZZ domain of p62, the Asp1721 in UBR4$^{UBR}$ corresponds to the Asp153 of UBR1 and UBR2 and Asp129 of p62[44,62]. Previous studies have shown that the Asp153 of UBR1 and UBR2 and Asp129 of p62 are critical for binding with ligands of the N-terminal residue[44,62]. The D153A of UBR1 and UBR2 and D129A of p62 mutant were reported to completely abolish the binding affinity for ligands in not only UBR1 and UBR2 but also p62/SQSTM1[48,63]. According to previous mutational analysis, we measured the binding affinity between UBR4$^{UBR}$ (D1721A), in which Asp1721 was substituted with Ala, and YEFS or REFS using ITC. The binding affinities between the two tetrapeptides and the D1721A mutant were reduced but were still retained for YEFS with a $K_D$ of $38.6 \pm 8.5$ μM, and showed almost the same binding affinity as the wild-type for REFS with a $K_D$ of $44.9 \pm 12.1$ μM (Fig. 6a, b and Supplementary Fig. 7a). In UBR4, Asp1721 did not play a role in the binding of the N-terminal residue of the ligand, especially arginine, even with YEFS.

We introduced F1671A and F1713A mutations and performed ITC to investigate whether the hydrophobic surface formed by the two phenylalanines is important for binding. We confirmed that UBR4$^{UBR}$ (F1671A) and UBR4$^{UBR}$ (F1713A) did not bind to YEFS or REFS (Fig. 6a, b and Supplementary Fig. 7b), indicating that these two phenylalanines are crucial for binding to the N-terminal residue of the ligand.

**Selective binding between the N-terminal residue of the ligand and UBR4$^{UBR}$ is mainly due to pi-interactions**. Based on the results of the TSA experiments, we could not explain why there was a difference of more than two-fold in the binding affinity between WEFS and LEFS, or why REFS, with its highly positively-charged arginine, had a higher binding affinity than LEFS, with its highly hydrophobic leucine (Fig. 2c). We already confirmed from the structure that the N-terminal residue, tyrosine, of the ligand is oriented toward the two phenylalanine residues of UBR4$^{UBR}$ (Fig. 5a). We calculated the distances and angles between these three aromatic rings and found the possibility of pi-interaction between them.

The pi-interaction is one of the intermolecular interactions, in which pi-electron clouds of aromatic rings interact with other aromatic rings, positive residues, and metals. There are parallel, intermediate, and T-shaped conformations in the pi–pi interactions between aromatic rings, which can be classified based on the distance between the centers of the aromatic rings ($R_{cen}$) and the angle ($\gamma$) between the surface normals perpendicular (Fig. 5b)[64–68].

Between Phe1671 and Phe1713 of the YIFS-UBR4$^{UBR}$ and Tyr1 of the ligand form trimeric pi–pi interactions, and Phe1671 and Phe3 of the ligand form typical T-shaped pi–pi interaction (Fig. 5c). In these pi–pi interactions, the distances ($R_{cen}$) fall within the range of 4.7 to 6.5 Å, which is the typical distance range for the pi–pi interaction. The angles ($\gamma$) between the surface normals of the two aromatic ring planes also fall within the pi–pi interaction range. Among these, the $\gamma$ between Phe1713 and Tyr1 (50.8°) and between Phe1671 and Phe3 (77.6°) was included in the T-shaped conformation, whereas the $\gamma$ between Phe1671 and Tyr1 (47.6°) and between Phe1671 and Phe1713 (44.9°) was included in the intermediate conformation (Fig. 5d). The presence of pi–pi interactions can greatly enhance the preference for amino acids containing aromatic rings, such as tyrosine.

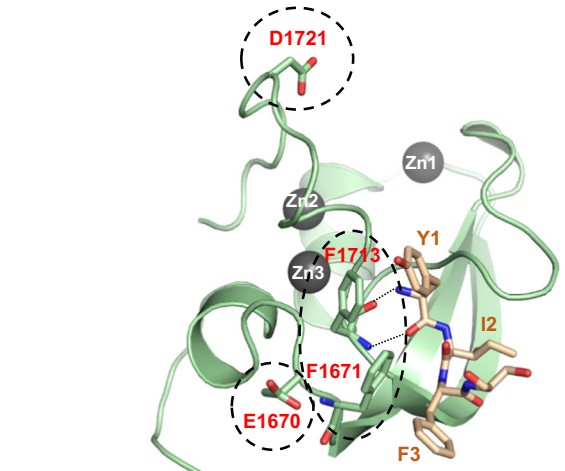

**b**

| Mutation | Peptide | $K_D$ (uM) |
|---|---|---|
| D1721A | YEFS | 38.6 ± 8.5 |
| | REFS | 44.9 ± 12.1 |
| E1670A | YEFS | 19.9 ± 0.8 |
| | REFS | 52.0 ± 3.5 |
| F1671A | YEFS | 208 ± 62 |
| | REFS | 660 ± 451 |
| F1713A | YEFS | No binding |
| | REFS | No binding |
| F1671I | YEFS | No binding |
| | REFS | No binding |
| F1713I | YEFS | No binding |
| | REFS | No binding |

\* Limit of detection : 50uM

**Fig. 6 The positions of mutations for validating pi-interactions and binding affinity between mutants and peptides. a** The mutant residues expected to be involved in the binding of ligand YIFS to UBR4$^{UBR}$ is presented in the structure. Mutant residues are indicated with a dotted circle. **b** UBR4$^{UBR}$ mutant constructs (D1721A, F1671A, F1713A, F1671I, F1713I, or E1670) with YEFS and REFS of the dissociation constant determined using isothermal titration calorimetry.

To determine the importance of pi-interactions in binding, we created F1671I (UBR4$^{UBR}$ (F1671I)) and F1713I (UBR4$^{UBR}$ (F1713I)) mutants by introducing isoleucine, which has almost the same hydrophobicity as phenylalanine, and measured the binding affinity of YEFS and REFS peptides with these mutants using ITC[69]. No binding between the two FI mutants and REFS or YEFS peptides was observed (Fig. 6a, b and Supplementary Fig. 7c). Thus, pi-interaction between the two phenylalanines is the major factor for binding and selectivity with the N-terminal residue of the ligand.

The results of ΔTm measurement for XEFS showed that tryptophan, tyrosine, phenylalanine, arginine, and histidine had higher binding affinities with UBR4$^{UBR}$ than with the others (Fig. 2c). All these amino acids have side chains capable of pi–pi interactions with phenylalanine. Based on our findings, pi–pi interaction is a major factor determining the binding affinity between the ligand and UBR4$^{UBR}$. Phe1671 and Phe1713 in YIFS-UBR4$^{UBR}$ are involved in binding through pi–pi interaction with Tyr1 and Phe3 of the ligand (Fig. 5c). Based on these results, it

can be inferred that residues with aromatic rings may also have high binding affinities because they can form pi–pi interactions with Phe1671 and Phe1713. Of the two phenylalanine residues crucial for type-2 N-degron binding, Phe1713 is fairly conserved in other UBR boxes, whereas Phe1671 is unique to UBR4 and is not conserved in other UBR boxes (Fig. 5e). The presence of two phenylalanine residues explains the strong binding affinity for type-2 N-degrons with an aromatic ring, and our observations indicate that Phe1671 plays an important role in binding specificity.

In the case of arginine, the guanidine group may form a pi-cation interaction with the benzene rings of phenylalanines[66–68]. We further examined the structure of UBR4$^{UBR}$ to assess the presence of another charge interaction with N-terminal arginine residue. We observed that E1670 is located close to two phenylalanines, F1671 and F1713. To verify its involvements in the charge interaction with N-terminal arginine, we created E1670A, which is mutated of Glu to Ala, and measured the binding affinity with REFS using ITC and YEFS, which served as a negative control. There was no difference in the binding affinities with the wild-type showed E1670A mutant of UBR4$^{UBR}$ for REFS with a $K_D$ of $52.0 \pm 3.5\,\mu M$ (Fig. 6a, b and Supplementary Fig. 7d). Also, E1670A mutant of UBR4$^{UBR}$ for YEFS showed similar binding affinity with wild-type, a $K_D$ of $19.9 \pm 0.8\,\mu M$ (Fig. 6a, b and Supplementary Fig. 7d). Therefore, we confirmed that UBR4$^{UBR}$ recognizes N-terminal arginine not via charge interaction but rather through a pi-cation interaction.

For histidine, the CH or NH of the imidazole ring can generally have a strong pi-interaction when facing the center of the benzene ring[70,71]. However, in the UBR4$^{UBR}$ structure, the benzene rings of phenylalanine are perpendicular to the imidazole ring, indicating a relatively lower affinity than the others.

**Structural differences among UBR4$^{UBR}$ and the UBR boxes of UBR1 and UBR2 lead to differences in selectivity for the second residues of ligands.** To investigate the differences in the binding affinity of UBR with the variation in the second residue of YXFS, we conducted TSA experiments and made some interesting observations (Fig. 2a). Small amino acid residues, such as alanine, serine, and glycine, were the best in binding to the second pocket. Interestingly, the long and hydrophobic methionine residue also bound well. In contrast, other hydrophobic residues, such as leucine, isoleucine, and phenylalanine, did not bind well. In addition, although the binding ability of glutamate is high, that of another negatively charged amino acid, aspartate, is not (Fig. 2a). To understand these differences, we used the Rosetta relaxation program[72–78], along with three water molecules anchored to the second pocket, to predict the structure of UBR4$^{UBR}$ complexed with YXFS peptides containing different residues at the second position.

Based on the predicted structures, it was observed that small residues could be accommodated in the small and hydrophobic secondary pocket of UBR4$^{UBR}$ (Fig. 7a, b). This pocket can only accommodate the Cα and Cβ carbons (Fig. 7c). The Cα of glycine, as well as the Cα and Cβ of serine and alanine, can bind to the pocket without any restrictions. In the case of methionine, despite the presence of a long hydrophobic chain, its Cα and Cβ appear to be capable of binding. However, the Cβ of isoleucine, leucine, and valine could not adhere to the second pocket because of the presence of Cγ, resulting in these residues exhibiting a lower affinity than other hydrophobic amino acids (Fig. 7c).

Glutamate and aspartate, which possess similar properties, were compared to further understand the differences in their binding affinities (Fig. 2a, b). Compared with glutamate, aspartate has a structure similar to that of leucine because it has one less

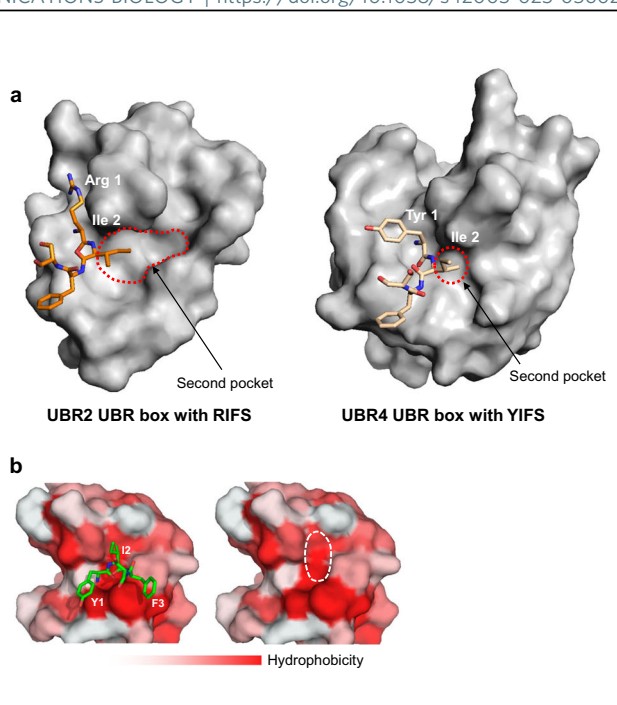

**a**

UBR2 UBR box with RIFS    UBR4 UBR box with YIFS

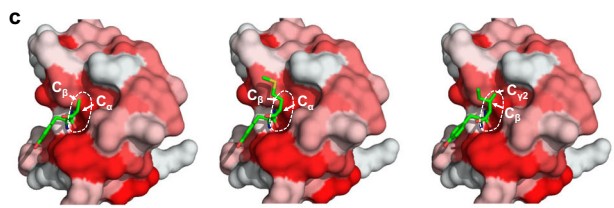

**b**

Hydrophobicity

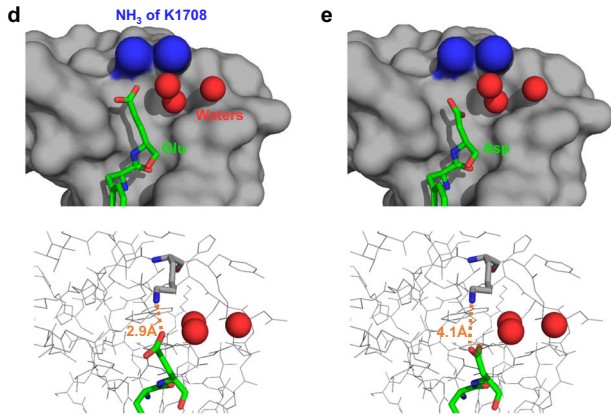

**c**

**d**    NH₃ of K1708    **e**

Waters    Glu    Asp

2.9Å    4.1Å

**f**

| Mutation | Peptide | $K_D$ (uM) |
|---|---|---|
| Wild Type | YEFS | 18.3 ± 0.9 |
| | YMFS | 46.1 ± 5.3 |
| | YKFS | No binding |
| | YDFS | No binding |
| K1708A | YEFS | 13.5 ± 2.6 |
| | YKFS | 33.3 ± 3.2 |
| | YDFS | No binding |

* Limit of detection : 50uM

carbon. Owing to steric hindrance caused by the carboxyl group, it becomes difficult for Cβ to stably adhere to the second pocket (Fig. 7c). Moreover, in the crystal structure, three water molecules were observed near Lys1708 in the second pocket (Fig. 7d, e). Because of the positioning of these water molecules, the side chain of Lys1708 is restricted to two possible conformations

**Fig. 7 Structural analysis of selectivity in the second pocket. a** The structures of ligand-bound UBR boxes of UBR2 and UBR4 represented as a surface. The second pocket of UBR4 is significantly smaller than that of UBR2. **b** YIFS-UBR4$^{UBR}$ with ligand is represented as a surface colored for hydrophobicity. Red indicates increasing hydrophobicity. **c** Models of the binding between UBR4$^{UBR}$ and hydrophobic ligands, with the surface colored according to hydrophobicity. The positions of Cα and Cβ atoms of the ligands are indicated. **d, e** Two possible locations for Lys1708 (blue) and three water molecules (red balls) bound to the second pocket are shown. Figures d and e represent models where the second residue of the ligand is Glu and Asp, respectively. The distance between the carboxyl group of each second residue and the primary amine group of Lys1708 is indicated at the bottom. **f** The dissociation constant ($K_D$) values measured through isothermal titration calorimetry for four peptides with UBR4$^{UBR}$ (WT) and three peptides with UBR4$^{UBR}$ (K1708A), respectively.

(Fig. 7d, e). Because the carboxyl group of glutamate can be positioned at a distance suitable for hydrogen bonding with the primary amine of Lys1708 (2.9 Å), it is expected to show strong binding affinity (Fig. 7d). The carboxyl group of aspartate was farther away from the primary amine of Lys1708 by one C-C distance compared to that of glutamate (4.1 Å) (Fig. 7e). As a result, hydrogen bonding became impossible, and YDSF exhibited almost no binding (Fig. 7f).

To verify our hypothesis based on this model, we introduced a K1708A mutation into UBR4$^{UBR}$ (UBR4$^{UBR}$ (K1708A)) and measured its binding affinities with tetrapeptides of YXFS using ITC. First, YDFS did not bind to UBR4$^{UBR}$ (WT) or UBR4$^{UBR}$ (K1708A). On the contrary, YEFS showed binding with both UBR4$^{UBR}$ (WT) and UBR4$^{UBR}$ (K1708A), with binding affinities of 18.3 ± 0.9 and 13.5 ± 2.6 μM, respectively. Compared to the ITC results for YEFS and wild-type UBR4$^{UBR}$, the binding affinity of YEFS with UBR4$^{UBR}$ (K1708A) slightly increased, and ΔH decreased (Fig. 7f and Supplementary Fig. 8). Substitution of Lys1708 with Ala made the charge interaction between the carboxyl group of the second glutamate of the ligand and the amine group of Lys1708 impossible. However, water molecules bound to the protein surface through the amine group of Lys1708 may have weakened or dissociated, resulting in a decrease in ΔH and ΔG, which lead to a slight increase in binding affinity.

The TSA results, YMFS showed the fourth strongest binding affinity for UBR4$^{UBR}$ at a ligand concentration of 2 mM (Fig. 2a). This is because the methionine residue has a long chain without branches, allowing Cα and Cβ to bind to the second pocket of UBR4$^{UBR}$ without steric hindrance. However, YKFS did not show binding in the TSA (Fig. 2a). Although lysine has a long chain without branches, similar to methionine, the amine group at the end of its positive charge makes it unable to bind because of repulsion by the positive charge of Lys1708. To investigate whether YKFS could bind when the K1708A mutation was introduced into UBR4$^{UBR}$, the binding affinity was measured using ITC. YKFS, which showed no binding with UBR4$^{UBR}$ (WT), exhibited binding affinity for UBR4$^{UBR}$ (K1708A) with a $K_D$ of 33.3 ± 3.2 μM, which was similar to the $K_D$ for the binding of YMFS to UBR4$^{UBR}$ (WT) with a $K_D$ of 46.1 ± 5.3 μM (Fig. 7f and Supplementary Fig. 8). Thus, we confirmed that YKFS has a binding affinity similar to that of YMFS, if there is no interference from the positive charge of Lys1708.

In conclusion, the binding of the second residue of the ligand to the second pocket is determined by how well Cα and Cβ of the amino acid accommodate to the small hydrophobic pocket, and the selectivity for charge is determined by Lys1708.

## Discussion

Unlike other UBRs, UBR4$^{UBR}$ directly recognizes type-2 N-degrons as well as type-1 N-degron, arginine. We measured the binding affinities by testing 20 different tetrapeptides as the N-terminal residue using TSA and ITC. We demonstrated that residues with aromatic rings and arginine have a relatively high binding affinity (Fig. 2c). We determined the reason for these differences vis-à-vis other UBRs using structural analysis. The first pocket in the binding site recognizing the N-terminal amino group of the ligand is almost identical to the UBR boxes of UBR1 and UBR2. However, the structure that recognizes the N-terminal guanidine group of the ligand is completely different from those of UBR1 and UBR2. In the structures of the UBR boxes of UBR1 and UBR2, Asp153 recognizes the side chain of the N-terminal amino acid and provides selectivity for ligand binding (Fig. 4a). However, in our structure, Asp1721 of UBR4 was located away from the binding site (Fig. 4b, c); instead, the two phenylalanines, Phe1671 and Phe1713, were seen to participate in binding with the side chain of the ligand (Fig. 5a). The benzene rings of phenylalanines provide affinity and selectivity for the ligand through pi–pi or pi-cation interactions.

The second pocket of UBR4$^{UBR}$, which recognizes the second residue of the ligand, was relatively small compared to those of UBR1 and UBR2 (Fig. 7a). Additionally, we confirmed through binding affinity measurements that the second residue of the ligand exhibited a strong binding affinity when it consisted of small amino acids or glutamate (Fig. 2a). Because this small pocket allows only the Cα and Cβ of the second amino acid of the ligand to bind, small amino acids, such as glycine, alanine, and serine, are preferred for the second residue of the ligand (Fig. 7c). For YEFS, there is a Lys1708 residue at the end of the second pocket, which provides binding specificity (Fig. 7d). Because lysine is present at an appropriate distance to allow binding only with glutamate through hydrogen bonds, it should enhance the binding specificity for glutamate. Furthermore, lysine prevents the binding of positively charged amino acids, such as arginine and lysine.

This study reveals a new structural mechanism for the recognition of not only type-1 N-degrons, but also type-2 N-degrons in the UBR box. These results highlight the importance of investigating the structural basis of the N-degron pathway. Moreover, these findings will lead to more active research on the actual substrate of UBR4 and should potentially contribute to the development of new therapeutic strategies targeting this pathway.

## Methods

**Cloning and protein purification.** The DNA fragments encoding Human UBR4$^{UBR}$ (residues 1660–1729) and Human UBR4$^{UBR}$ mutants (F1671A, F1713A, F1671I, F1713I, D1721A, K1708A, or E1670A) were cloned into the 10H-MBP pHC plasmid fused, respectively, with N-terminal (His)$_{10}$-linked maltose-binding protein(10His-MBP) that was modified from the pET21a plasmid in our lab. The recombinant proteins were expressed in *Escherichia coli* BL21(DE3) RIL cells (Thermo Fischer Scientific, USA) cultured in Luria-Bertani medium (LB, Ambrothia) at 18 °C for 16 h after induction with 0.5 mM isopropyl-β-D-thiogalactoside (IPTG). *E. coli* cells expressing Human UBR4 were sonicated and centrifugated at 13000 rpm for 1 h. After centrifugation, the cleared cell supernatant was loaded onto a Ni-NTA resin (Cytiba, Sweden). Proteins were eluted with an elution buffer containing 50 mM Tris-HCl (pH 7.5), 200 mM NaCl, 3 mM β-mercaptoethanol (βME), and 300 mM imidazole. After Ni-NTA affinity chromatography, the N-terminal 10His-MBP-tag was digested by TEV treatment, and size-exclusion chromatography was performed using a HiLoad

26/600 Superdex 75 pg gel filtration column (Cytiva, USA). The final sample for crystallization was equilibrated with a buffer containing 50 mM Tris-HCl (pH 7.5), 200 mM NaCl, and 2 mM dithiothreitol (DTT). For preparation of ITC samples, 10His-MBP-tag Human UBR$^{UBR}$ proteins and 10His-MBP-tag Human UBR$^{UBR}$ mutant proteins were loaded onto Ni-NTA resin and subjected to size-exclusion chromatography, without TEV digestion. The final sample for ITC was equilibrated with a buffer containing 50 mM Tris-HCl (pH 7.5), 200 mM NaCl, and 5 mM β-mercaptoethanol (βME).

Human YIFS-UBR4 (residues 1660–1729) containing YIFS residues at the N-terminal site, was cloned into the LC3B-fusion plasmid with an N-terminal 10His-MBP. The recombinant protein was expressed in *E. coli* BL21(DE3) RIL cells cultured in LB at 18 °C for 16 h after induction with 0.5 mM IPTG. Human YIFS-UBR4 was loaded onto a Ni-NTA resin (Cytiba, Sweden). The proteins were eluted with an elution buffer containing 50 mM Tris-HCl (pH 7.5), 200 mM NaCl, 3 mM βME, and 300 mM imidazole. After Ni-NTA affinity chromatography, N-terminal 10His-MBP was digestion by treatment with human ATG4B protein, and size-exclusion chromatography was performed using a HiLoad 26/600 Superdex 75 pg gel filtration column (Cytiva, USA). The final sample was equilibrated in a buffer containing 50 mM Tris-HCl (pH 7.5), 200 mM NaCl, and 2 mM DTT.

**Crystallization, data collection, and refinement**. Purified UBR4$^{UBR}$ protein (3 mg/mL) was crystallized at 18 °C using the sitting-drop vapor diffusion method and by mixing an equal volume of protein with a reservoir solution containing 0.1 M DL-malic acid (pH 7.0) and 18% (w/v) polyethylene glycol 3350 (PEG3350). X-ray diffraction data were collected for the obtained crystals containing 17% glycerol as a cryoprotectant reagent. Diffraction data were collected on beamline 11 C using a Scan 4D system at the Pohang Accelerator Laboratory, Korea, and processed using the XDS Program package.

Purified YIFS-UBR4 protein (3 mg/mL) was crystallized at 18 °C using the sitting-drop vapor diffusion method and by mixing an equal volume of protein with a reservoir solution containing 0.1 M Bis-Tris (pH5.6), 29% (w/v) polyethylene glycol 3350 (PEG3350), 0.2 M lithium sulfate monohydrate. YIFS-UBR4 crystals were soaked in a reservoir solution containing 12% glycerol as a cryoprotectant. The diffraction data for UBR4$^{UBR}$ were collected at a wavelength of 1.283 Å on beamline 11 C (BL-11C) at the Pohang Accelerator Laboratory (PAL), Korea. The diffraction data for YIFS-UBR4$^{UBR}$ were collected at a wavelength of 1.282 Å on beamline 5 C (BL-5C), also at PAL, Korea. The collected data were processed using the XDS Program package.

The Protein structures were determined and refined using Phenix. In the UBR4$^{UBR}$ structure, 92.26 % of the amino acids are in favored region, 7.04 % are in allowed region, and there are no outliers. In the YIFS-UBR4$^{UBR}$ structure, 94.29 % of amino acids are in favored region, 5.71 % are in allowed region, and there are no outliers.

A more detailed information of data collection and refinement is provided in Table 1.

**Thermal shift assay**. UBR4 with different peptides was subjected to TSA using the Protein Thermal Shift Dye kit$^{TM}$ (Life Technologies). Each reaction mixture contained 20 µL of solution with 6.25 µL of 25 µM UBR4 or YIFS-UBR4, 6.25 µL of YXFS or XEFS peptide (0.03125 mM to 2 mM, dilution of 1/2 at the start of 2 mM concentration), 5 µL of Protein Thermal Shift$^{TM}$ buffer, and 2.5 µL of 1 X Protein Thermal Shift$^{TM}$ dye. Samples were incubated at 25 °C for 5 min before heating at temperatures from

**Table 1 Data collection and refinement statistics for UBR4$^{UBR}$ and YIFS-UBR4$^{UBR}$.**

|  | UBR4$^{UBR}$ | YIFS-UBR4$^{UBR}$ |
|---|---|---|
| Data collection |  |  |
| Space group | $P2_12_12_1$ | $P2_12_12_1$ |
| Cell dimensions |  |  |
| *a, b, c* (Å) | 43.67, 80.75, 81.32 | 36.58, 38.28, 55.4 |
| α, β, γ (°) | 90, 90, 90 | 90, 90, 90 |
| Resolution (Å) | 29.76-2.18 (2.258-2.18) | 26.45-1.65 (1.709-1.65) |
| $R_{sym}$ | 0.15 (1.397) | 0.09552 (0.2789) |
| $I/\sigma I$ | 13.99 (2.09) | 24.84 (8.45) |
| Completeness (%) | 98.04 (98.14) | 99.32 (98.64) |
| Redundancy | 12.7 (13.0) | 12.3 (10.4) |
| Refinement |  |  |
| Resolution (Å) | 2.18 | 1.65 |
| No. reflections | 15341 (1490) | 9760 (943) |
| $R_{work}$ / $R_{free}$ | 0.2192 / 0.2707 | 0.2345 / 0.2880 |
| No. atoms |  |  |
| Protein | 1689 | 592 |
| Ligand/ion | 9 (Zinc) | 3 (Zinc) |
| Water | 55 | 102 |
| *B*-factors |  |  |
| Protein | 51.70 | 16.83 |
| Ligand/ion | 59.95 (Zinc) | 12.95 (Zinc) |
| Water | 49.29 | 28.89 |
| R.m.s. deviations |  |  |
| Bond lengths (Å) | 0.008 | 0.008 |
| Bond angles (°) | 0.85 | 1.00 |

25 °C to 95 °C, increased at a rate of 0.2 °C per seconds. Finally, samples were incubated at 95 °C for 5 min. Fluorescence signals were monitored using a real-time PCR system. Data were analyzed using the Thermal Shift software (Life Technologies). All peptides were incubated alone for the no-protein control and gave flat lines at all temperatures. All proteins were incubated alone to determine the Tm of the UBR box of UBR4, and the obtained values were compared with the Tm of UBR4 with different peptide complexes. The maximum change in the Tm between the UBR box of UBR4 and its peptide complex indicated the highest binding affinity for that complex. To increase accuracy of results, thermal shift assays for each peptide were conducted in 3 replicates. Standard deviation (SD) which represented by error bars was calculated for each ΔTm measurement.

**Isothermal titration calorimetry**. All protein samples were obtained in a solution containing 50 mM Tris-HCl pH 7.5, 200 mM NaCl, and 5 mM βME. YXFS and XEFS peptides, where X represents one of the 20 natural amino acids, were prepared in a solution containing 50 mM Tris-HCl pH 7.5, 200 mM NaCl, and 5 mM βME, which was identical to the protein sample solution. The sample cell contained 1400 µL of 0.05 mM UBR4$^{UBR}$ and the injected samples comprised 300 µL of 1 mM YXFS or XEFS peptide. In the case of YIFS and RIFS peptides, to enhance weak binding affinities, 4.5 mM peptide was used for ITC studies. Similarly, UBR4$^{UBR}$ concentration was also increased 4.5 times to 0.225 mM. VDFS and YDFS peptides were used as negative controls, and these were also conducted to a 4.5 times of previous concentrations. For each reaction, 5 µL peptide was injected for 20 s. All measurements were performed at 25 °C on a VP-ITC microcalorimetry system (MicroCal). Data were analyzed using the Origin software (OriginLab Corp).

**Reporting summary**. Further information on research design is available in the Nature Portfolio Reporting Summary linked to this article.

## Data availability

The atomic coordinates and structure factors of UBR4$^{UBR}$ and YIFS-UBR4$^{UBR}$ have been deposited in the Protein Data Bank (PDB) under the accession codes 8J9Q and 8J9R, respectively. The ITC raw data can be found in Supplementary Data 1, and the TSA raw data can be accessed in Supplementary Data 2.

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

## Acknowledgements

We thank Prof. Hyun-Kyu Song (Korea University) for providing the LC3B tag system that allowed us to initiate the N-terminus of the protein with desired amino acids. The data used in this study were obtained from beamlines 5C and 11C at the Pohang Accelerator Laboratory in Korea. This work was supported by the National Research Council of Science and Technology (CRC22021-700), the KRIBB Research Initiative Program (KGM9952314), and the National Research Foundation of Korea (2021M3A9G8025599).

## Author contributions

D.E.J.: determination of protein structures and structural analysis, TSA and ITC assay, and manuscript writing. H.S.L.: ITC assay and ITC data analysis. B.K.: Data and structural analysis and data proofreading. C-H.K.: manuscript proofreading and writing advice. S.J.K.: As a co-corresponding author, experimental design, data analysis, and proofreading. H-C.S.: As a co-corresponding author, experimental design, data analysis, proofreading, data organization, manuscript writing, structural analysis, and modeling using Rosetta.

## Competing interests

The authors declare no competing interests.
