## [Peer review file · Communications Biology]

Reviewers' comments:

Reviewer #1 (Remarks to the Author):

This manuscript by Jeong et al. dissected the biochemical basis underlying the regulation of UBR4-mediated N-end rule substrate ubiquitylation. They have employed a variety of methods in analyzing the interaction between UBR4 E3 and N-degrons. The results revealed novel insights how UBR4 recognizes its substrates albeit the fragments of UBR4 are used. The conclusions are largely solid and supported by the data.

There are several specific issues:

1) The writing could be improved. For example, the first sentence of the abstract is rather long. And there are some typos, like "subfamily-,2" on page 9. More careful editing would help.

2) There are other structural studies done with N-degrons and UBRs, including yeast homologues. More comprehensive analysis/discussion would improve the paper.

3) Fig. 2c is discussed ahead of Fig. 2b.

4) Longer UBRs (i.e., yeast UBR1) have been used in the structural studies. It'd be important to discuss the differences and similarities of those studies with this one.

Reviewer #2 (Remarks to the Author):

This manuscript is the first to report structural and mechanistic analyses of how UBR4 without its N-domain binds to both type 1 and -2 N-degrons. The authors used well-controlled experiments - thermal shift assay, isothermal titration calorimetry, x-ray crystallography and molecular modelling to measure/analyze the binding affinity/mode of UBR4UBR with 20 different tetrapeptides. The authors also performed structural analyses to determine that the two important Phenylalanine residues participate in the ligand binding via pi-pi interactions. Overall, this study elegantly unravels a new structural mechanism for recognizing both type-1 and type-2 N-degrons by the UBR box, which provides novel insights in the field of the N-degron pathway. The authors also show clear expertise and grasp of the required techniques to draw insightful conclusions. Overall, this study is an excellent example of structural analyses of the N-degron pathway. I feel this manuscript should be accepted for publication if the following comments are addressed.

Major comment

1. It has been reported that pi interaction is important to substrate binding. This has been demonstrated through the measurement of binding affinity using ITC between mutated UBR boxes and substrates. However, in the case of the Arg/N-degron substrate which is very long and positive charged, there is a possibility that charge-charge interaction with acidic residues may be further away than the two phenylalanines of UBR4. The manuscript would be strengthened if the authors could investigate to see whether there are any other residues capable of binding with arginine. If the results are indeed as expected, it will be interesting to measure the binding affinity with appropriate mutants.

2. The authors should further clarify figures and figure legends with either in-picture legends, arrows, lines, etc or additional information to help emphasize their conclusions. For example, in figure 7b, it is not clear what the tetrapeptide is used for molecular modelling, and the white circle to show the UBR box pocket in which the tetrapeptide resides in is barely visible.

3. While not overt, the grammatical errors and awkward sentence structures in the manuscript hamper readability. The manuscript should be editorially revised to improve readability.

Minor comments

1. It is recommended to add an explanation of the LC3B tag system, potentially accompanied by graphical illustration.

2. "N-terminal residue" is an official nomenclature in this field. The authors would be better off in using "N-terminal residue" instead of "first residue" in the sentence.

3. In the first few sentences on page 6, it is recommended for the authors to re-phrase some of the wording so that they clearly indicate how type 2 N-degrons are not only recognized by the N-domain of UBR1 and UBR2 but also by UBR4 without the N-domain through an unknown mechanism. It would be better to remove the word "only" in line 2.

4. Figure 3b is not mentioned in the manuscript.

5. (Page 11) "The binding affinities...", there is a typo - "reducd".

6. (Page 11) In "In previous studies, this mutation...", the authors should include a specific domain of p62 for clarify and add an explanation of the ZZ domain and how it relates to the UBR box for sake of clarity.

8. (Page 13) In "All these amino acids...", "gindings" should be "findings."

9. (Page 14) In "In addition, although...", "Fig. 2A" should be "Fig. 2a".

10. (Page 14) In "Compared with glutamate...", "smilar" should be "similar".

11. (Page 17) In "We measured...", "varying 20 tetrapeptides" could be replaced with "by testing 20 different tetrapeptides".

12. (Page 17) In "However, in our structure...", "Phe1617" should be "Phe1671".

13. (Page 18) In "These results highlight...", "N-end rule pathway" could be changed to "N-degron pathway", the latter now being formally accepted in the field.

Reviewer #3 (Remarks to the Author):

This is a potentially interesting manuscript. UBR4 is an important protein and there is little information about the function and specificity of its UBR-box. Unfortunately, the manuscript suffers

from many, many deficiencies that preclude assessment of the work.

Specific issues:

The quality of ITC data is currently unpublishable. It is unclear that the results in Fig 1 actually represent binding. The vertical scale is 10x larger than the more reasonable looking ITCs in the supplementary material. If the vertical scale in Supp Fig 1C (YDFS) were similarly expanded, it would look like Fig 1A or B. The authors are inconsistent in whether this represents binding or not.

The authors should redo the measurements in a buffer such as HEPES. Quoting reference 54 (Munoz-Escobar et al), "the use of 50 mM Tris as a buffer had an impact on the affinities measured previously. We repeated ITC experiments using 20 mM HEPES buffer and observed markedly higher affinities. For the highest-affinity peptides, the improvement in affinity was roughly 40-fold." This has the potential of significantly improving the quality of ITC data.

If the buffer change doesn't help, then ITC experiments need to be repeated with higher protein and peptide concentrations. The crystallization was done at 3 mg/ml. ITC experiments need to be done at a similar concentration so that the ratio of [protein] / K_d = 10 or more.

The TSA melting curves should be included as supplementary material.

Consider using NMR spectroscopy to characterize tetrapeptide binding. NMR is excellent for measuring weak binding.

The authors need to include the correct PDB validation reports (intended for manuscript review).

No information is provided about how the structures were phased. The locations of the Zn atoms should be confirmed by anomalous signal.

The authors need to investigate and correct the cause of the high R-factors in the YIFS-UBR4 structure. If they can't be improved, they need to include the raw data and pdb file for review. Why was the resolution cut off at 1.65 Å in this dataset?

Density of an omit map of the bound peptide should be shown as a supplementary figure.

Minor points

Put the supplementary figure legends in the same document as the supplementary figures.

"Not determined" in the ITC tables suggests the measurement was not done. If the authors mean that the affinity was too weak to be quantified, they should write "No binding" and specify the limit of detection, e.g. 0.5 mM.

Include page numbers

Point-by-Point Responses

Reviewers' comments:

Reviewer #1

1. The writing could be improved. For example, the first sentence of the abstract is rather long. And there are some typos, like "subfamily-,2" on page 9. More careful editing would help.

→ I have corrected as many typos as possible in the manuscript, and I have received editing assistance from an editor. I have attached the editing certificate.

2. There are other structural studies done with N-degrons and UBRs, including yeast homologue. More comprehensive analysis/discussion would improve the paper.

→ I have included a comparison figures of the yeast UBR1 UBR box in Figure 3b, 4a, and 4c, and I have also added it to the manuscript on page 9.

3. Fig. 2c is discussed ahead of Fig. 2b.

→ I have rearranged the order of the data in Figure 2.

4. Longer UBRs (i.e., yeast UBR1) have been used in the structural studies. It'd be important to discuss the differences and similarities of those studies with this one.

→ I have added a figure to the supplementary information 6 that compares the cryo-EM structure of full length yeast UBR1. I have also added the relevant content to the manuscript on page 9.

Reviewer #2

Major comment

1. It has been reported that pi interaction is important to substrate binding. This has been demonstrated through the measurement of binding affinity using ITC between mutated UBR boxes and substrates. However, in the case of the Arg/N-degron substrate which is

very long and positive charged, there is a possibility that charge-charge interaction with acidic residues may be further away than the two phenylalanines of UBR4. The manuscript would be strengthened if the authors could investigate to see whether there are any other residues capable of binding with arginine. If the results are indeed as expected, it will be interesting to measure the binding affinity with appropriate mutants.

→ Upon closer examination of the structure, it was observed that there is a presence of glutamate near the binding site of the ligand. We substituted this glutamate to alanine and measured the binding affinity with YEFS and REFS through ITC. However, the results did not show any significant changes. No other residues that could potentially affect the binding of arginine were identified in other regions. I mentioned these ITC data to the Figure 6b and supplementary information 8d. I have also added the relevant content to the manuscript on page 15 and 16.

2. The authors should further clarify figures and figure legends with either in-picture legends, arrows, lines, etc or additional information to help emphasize their conclusions. For example, in figure 7b, it is not clear what the tetrapeptide is used for molecular modelling, and the white circle to show the UBR box pocket in which the tetrapeptide resides in is barely visible.

→ I have corrected accordingly in manuscript.

3. While not overt, the grammatical errors and awkward sentence structures in the manuscript hamper readability. The manuscript should be editorially revised to improve readability.

→ I have corrected as many typos as possible in the manuscript, and I have received editing assistance from an editor. I have attached the editing certificate.

Minor comments

1. It is recommended to add an explanation of the LC3B tag system, potentially accompanied by graphical illustration.

→ I have added a figure to the supplementary information 7a. I have also added the relevant content to the manuscript on page 12.

2. “N-terminal residue” is an official nomenclature in this field. The authors would be better off in using “N-terminal residue” instead of “first residue” in the sentence.

→ I have corrected accordingly in manuscript.

3. In the first few sentences on page 6, it is recommended for the authors to re-phrase some of the wording so that they clearly indicate how type 2 N-degrons are not only recognized by the N-domain of UBR1 and UBR2 but also by UBR4 without the N-domain through an unknown mechanism. It would be better to remove the word “only” in line 2.

→ I have corrected accordingly in manuscript.

4. Figure 3b is not mentioned in the manuscript.

→ I have incorporated the additions on page 10.

5. (Page 11) “The binding affinities...”, there is a typo - “reduced”.

→ I have corrected accordingly in manuscript.

6. (Page 11) In “In previous studies, this mutation...”, the authors should include a specific domain of p62 for clarify and add an explanation of the ZZ domain and how it relates to the UBR box for sake of clarity.

→ I have added accordingly in manuscript page 12.

8. (Page 13) In “All these amino acids...”, “gindings” should be “findings.”

→ I have corrected accordingly in manuscript.

9. (Page 14) In “In addition, although...”, “Fig. 2A” should be “Fig. 2a”.

→ I have corrected accordingly in manuscript.

10. (Page 14) In “Compared with glutamate...”, “smilar” should be “similar”.

→ I have corrected accordingly in manuscript.

11. (Page 17) In “We measured...”, “varying 20 tetrapeptides” could be replaced with “by testing 20 different tetrapeptides.”

→ I have corrected accordingly in manuscript.

12. (Page 17) In “However, in our structure...”, “Phe1617” should be “Phe1671”.

→ I have corrected accordingly in manuscript.

13. (Page 18) In “These results highlight...”, “N-end rule pathway” could be changed to “N-degron pathway”, the latter now being formally accepted in the field.

→ I have corrected accordingly in manuscript.

Reviewer #3

1. The quality of ITC data is currently unpublishable. It is unclear that the results in Fig 1 actually represent binding. The vertical scale is 10x larger than the more reasonable looking ITCs in the supplementary material. If the vertical scale in Supp Fig 1C (YDFS) were similarly expanded, it would look like Fig 1A or B. The authors are inconsistent in whether this represents binding or not.

The authors should redo the measurements in a buffer such as HEPES. Quoting reference 54 (Munoz-Escobar et al), "the use of 50 mM Tris as a buffer had an impact on the affinities measured previously. We repeated ITC experiments using 20 mM HEPES buffer and observed markedly higher affinities. For the highest-affinity peptides, the improvement in affinity was roughly 40-fold." This has the potential of significantly improving the quality of ITC data.

If the buffer change doesn't help, then ITC experiments need to be repeated with higher protein and peptide concentrations. The crystallization was done at 3 mg/ml. ITC experiments need to be done at a similar concentration so that the ratio of [protein] / K_d = 10 or more.

→ In order to obtain more favorable experimental results, we performed ITC assay using wild-type UBR4 UBR box under 50 mM HEPES buffer condition (pH 7.5). However, we were unable to achieve better results for the majority of samples. Surprisingly, even YMFS, which showed binding in both TSA and ITC data in Tris buffer condition, exhibited almost no binding under HEPES buffer conditions. ITC data under HEPES buffer conditions are summarized in the attached file.

The range of K_D values attainable through ITC is dependent on the concentration of the protein in the cell of ITC machine, and in our experiments, it was 50 μM . Also, the profile of the curve is determined by the c -value, the $c = nM/K_D$. Stoichiometry (n) of our sample is 1 and M is 50 μM . The c -value must be between 1 and 1000. Therefore, K_D range of our ITC condition is from 50 nM to 50 μM . Accordingly, in Figure 1, the weak binding of the two peptides, YIFS and RIFS, to UBR4 UBR box made it challenging to accurately determine the affinity values through ITC. Fortunately, based on the reviewer's advice, we performed ITC by increasing the concentrations of both the protein and the peptide by 4.5-fold. As a result, we were able to determine the affinities as $385.13 \mu\text{M} \pm 59.51$ and $626.35 \pm 29.20 \mu\text{M}$, respectively. However, ITC assay were not performed for the other peptides in high concentration of samples. Firstly, at this concentration, a substantial amount of materials is required, such as 20 L of E.coli and 1.2 mg of peptide, per single measurement. Moreover, while it is appropriate to determine the affinity of peptides that are considered real substrates, evaluating the binding affinity with peptides of very low affinity, such as YDFS, which has a low likelihood of actual binding, holds limited significance in practice. Expanding the vertical scale of the ITC data for YDFS reveals a curve that can be interpreted as having extremely weak binding. When measured at a 4.5-fold higher concentration, a K_D value of $123.55 \pm 5.5 \mu\text{M}$ was obtained (refer to the figure below). However, it is difficult to ascribe biological meaning to this result.

The ITC measurements of YIFS and RIFS with UBR4 UBR box in Figure 1 were conducted without knowing the real substrate of UBR4. Through this, we gained an experimental insight that the N-terminal residue, unlike other UBR boxes where it is typically arginine, might be tyrosine instead and could potentially exhibit better affinity. Therefore, we have added three ITC data in the manuscript for these two experiments and the negative control, VIFS, measured at a 4.5-fold higher concentration. Additionally, The reason we could only increase the protein concentration by up to 4.5-fold was due to the stability of the protein.

2. The TSA melting curves should be included as supplementary material.

→ I have added a figure to the supplementary information 2 and 4.

3. Consider using NMR spectroscopy to characterize tetrapeptide binding. NMR is excellent for measuring weak binding.

→ Our laboratory lacks experience in determining protein structures using NMR, and the conditions for performing NMR assay are not feasible for us. Furthermore, the affinity between REFS and UBR4 UBR box is relatively weak, at approximately $40 \mu\text{M}$. To observe this binding through NMR experiments, the concentration of REFS would need to be increased by over 40-fold compared to the protein concentration. However, under this condition, the signal interference caused by the REFS peptide will be able to so significant that determining the structure becomes unfeasible.

4. The authors need to include the correct PDB validation reports (intended for manuscript review).

→ I sent cover letter with the PDB validation reports for manuscript review file.

7. No information is provided about how the structures were phased. The locations of the zinc atoms should be confirmed by anomalous signal.

→ Our structures were phased with zinc ions. The data have been summarized and included in the statistics table. The position of the zinc ions have been confirmed through anomalous signal.

8. The authors need to investigate and correct the cause of the high R-factors in the YIFS-UBR4 structure. If they can't be improved, they need to include the raw data and pdb file for review. Why was the resolution cut off at 1.65 Å in this dataset?

→ The high R-factor seems to have been influenced by strong noise from the anomalous signal of zinc. When I received data with a wavelength of 1 Å and determined the structure by molecular replacement, the R-factor did not go below 30 % during refinement. However, after receiving data with a wavelength of 1.282 Å and performing refinement with the inclusion of the anomalous signal, I could reduce the R-factor to its current value. The refined structure shows no anomalies, and water molecules are also well positioned. To demonstrate this, we have attached our raw data and the PDB file. The dataset's cutoff of 1.65 Å is due to the fact that with a wavelength of 1.282 Å from PAL-BL-5C, the maximum achievable resolution is 1.65 Å.

9. Density of an omit map of the bound peptide should be shown as a supplementary figure.

→ I have added a figure to the supplementary information 7c.

Minor points

1. Put the supplementary figure legends in the same document as the supplementary figures.

→ I have corrected accordingly in manuscript.

2. "Not determined" in the ITC tables suggests the measurement was not done. If the authors mean that the affinity was too weak to be quantified, they should write "No binding" and specify the limit of detection, e.g. 0.5 mM.

→ I have corrected accordingly in manuscript.

3. Include page numbers

→ I have corrected accordingly in manuscript.

REVIEWERS' COMMENTS:

Reviewer #2 (Remarks to the Author):

All of my comments were sufficiently addressed. I have no further concerns and feel the manuscript is now publishable. This work will be an important contribution to the field.

Reviewer #3 (Remarks to the Author):

For clarity, the previous review didn't suggest structure determination by NMR was necessary. Rather, it was a suggestion that NMR be used for studying the weak binding ligands. NMR is superior to ITC for estimating binding affinities of weakly binding ligands. When using isotopically labeled protein, the presence of a high concentration of ligand does not interfere with the protein NMR spectrum. The ligand concentration only needs to be several-fold higher than the K_d to get an accurate measurement.

There are a few changes to be made before the final acceptance of the manuscript:

1. When performed at a higher protein concentration, the YDFS peptide bound with K_d of 123 μM (figure in the rebuttal letter). I agree that it might be difficult to ascribe biological meaning to this binding, but its higher affinity than the YIFS and RIFS peptides (385 and 626 μM) and should be reported. The revised manuscript misleadingly claims no binding for this peptide in Figures 2b and 7f. The YIFS data should be added either to Figure 1 or as a supplemental figure.

2. Too many significant figures are reported in ITC affinities. For example, line 112 on page 6, " $K_D = 626.35 \pm 29.20 \mu\text{M}$ " should be changed to " $K_D = 630 \pm 60 \mu\text{M}$ " (at best). The authors are confusing the error estimates from the fitting with the actual experimental error and overlooking other sources of error such as concentration measurements which are typically at least 10%. (While the errors from the fittings add to the overall uncertainty, they do not decrease it.) Furthermore, since the affinities are not measured to a precision of 3 figures, there is no point in including the extra digits when reporting the values. As a check of the accuracy of the affinities, the authors can look at the stoichiometries. In Figure 1, the stoichiometry N of YIFS is reported as 1.63. This means that the K_D can't be more precise than $\pm 60\%$.

3. In the ITC figures, please remove x4.5 label and report the experimental protein and peptide concentrations in the figure or figure legend.

Point-by-Point Responses

Reviewers' comments:

Reviewer #3

1. When performed at a higher protein concentration, the YDFS peptide bound with K_d of 123 μM (figure in the rebuttal letter). I agree that it might be difficult to ascribe biological meaning to this binding, but its higher affinity than the YIFS and RIFS peptides (385 and 626 μM) and should be reported. The revised manuscript misleadingly claims no binding for this peptide in Figures 2b and 7f. The YIFS data should be added either to Figure 1 or as a supplemental figure.

→ I have corrected accordingly in supplemental figure 1b and have also added it to the manuscript on page 8 from line 8 to 12.

2. Too many significant figures are reported in ITC affinities. For example, line 112 on page 6, " $K_D = 626.35 \pm 29.20 \mu\text{M}$ " should be changed to " $K_D = 630 \pm 60 \mu\text{M}$ " (at best). The authors are confusing the error estimates from the fitting with the actual experimental error and overlooking other sources of error such as concentration measurements which are typically at least 10%. (While the errors from the fittings add to the overall uncertainty, they do not decrease it.) Furthermore, since the affinities are not measured to a precision of 3 figures, there is no point in including the extra digits when reporting the values. As a check of the accuracy of the affinities, the authors can look at the stoichiometries. In Figure 1, the stoichiometry N of YIFS is reported as 1.63. This means that the K_D can't be more precise than $\pm 60\%$.

→ I have corrected accordingly in manuscript on page 6 line 19 and line 20, page 8 line 4, page 8 line 10, page 9 line 4, page 9 line 7, page 13 line 17, page 13 line 18, page 16 line 12, page 16 line 14, page 18 line 14 and page 19 from line 7 to line 9.

I have also corrected accordingly in figure 1c, 2b, 2d, 6b, 7f and supplementary figure 1b.

3. In the ITC figures, please remove x4.5 label and report the experimental protein and peptide concentrations in the figure or figure legend.

→ I have corrected accordingly in figure 1 and its figure legend.